Association of endothelial dysfunction with sarcopenia and muscle function in a relatively young cohort of kidney transplant recipients

Khoo Siok-Bin 1
Lin Yu-Li 1 2
Ho Guan-Jin 3
http://orcid.org/0000-0002-0321-910X Lee Ming-Che 1 3 4 mclee1229@mail.tcu.edu.tw
Hsu Bang-Gee 1 2 gee.lily@msa.hinet.net
1 School of Medicine, Tzu Chi University , Hualien , Taiwan
2 Division of Nephrology, Hualien Tzu Chi Hospital, Buddhist Tzu Chi Medical Foundation , Hualien , Taiwan
3 Department of Surgery, Hualien Tzu Chi Hospital, Buddhist Tzu Chi Medical Foundation , Hualien , Taiwan
4 Department of Surgery, Wan Fang Hospital, Taipei Medical University , Taipei , Taiwan
Smith Edward
Electronic publication date: 2021 Nov 22
Publication date: 2021
Volume: 9
Electronic Location ID: e12521
Received 2021 Jun 15; Accepted 2021 Oct 29
Copyright: © 2021 Khoo et al.
Copyright year: 2021
Copyright holder: Khoo et al.
License: This is an open access article distributed under the terms of the Creative Commons Attribution License, which permits unrestricted use, distribution, reproduction and adaptation in any medium and for any purpose provided that it is properly attributed. For attribution, the original author(s), title, publication source (PeerJ) and either DOI or URL of the article must be cited.
License URL: https://creativecommons.org/licenses/by/4.0/

Keywords: Endothelial function, Sarcopenia, Muscle mass, Muscle strength, Gait speed, Kidney transplantation

Funding: Buddhist Tzu Chi General Hospital, Hualien, Taiwan TCRD108-53 This study was supported by a grant from Buddhist Tzu Chi General Hospital, Hualien, Taiwan under Grant TCRD108-53. The funders had no role in study design, data collection and analysis, decision to publish, or preparation of the manuscript.

==============================
Background

Sarcopenia and endothelial dysfunction are both common among kidney transplant (KT) recipients. We aimed to evaluate the association between endothelial dysfunction and sarcopenia, as well as its individual components.

Methods

Vascular reactivity index (VRI), skeletal muscle index (SMI = skeletal muscle mass/height2), handgrip strength (HGS), and 6-meter usual gait speed (GS) were measured in 95 KT recipients. Low SMI was defined as SMI less than 10% of the sex-specific reference values from Chinese adults; low HGS as HGS < 28 kg for men and < 18 kg for women; slow GS as GS below 1.0 m/s. Sarcopenia was diagnosed based on the presence of low SMI as an essential criterion, accompanied by either low HGS or slow GS. Vascular reactivity was classified as being indicative of poor (VRI < 1.0), intermediate (1.0 ≤ VRI < 2.0), or good (VRI ≥ 2.0) vascular reactivity.

Results

Of the 95 patients, aged 45.2 ± 10.9 years, 11.6% had sarcopenia and 13.7% had poor vascular reactivity. Patients with sarcopenia were lower in body mass index (p = 0.001) and VRI (p = 0.041), and have a higher proportion of low muscle mass (p < 0.001), low HGS (p < 0.001), and slow GS (p = 0.001). Patients with poor vascular reactivity have a higher proportion of sarcopenia (p = 0.005), low HGS (p = 0.006), and slow GS (p = 0.029). Multivariate logistic regression analysis showed that patients in the poor VRI group were significantly associated with sarcopenia (odds ratio, OR = 6.17; 95% confidence interval [1.06–36.04]; p = 0.043), comparing to those with good VRI. We further analysed the effects of VRI on individual components of sarcopenia and found that VRI predicted slow GS significantly (OR = 0.41; 95% CI = [0.21–0.79]; p = 0.007), but not low SMI (OR = 1.15; 95% CI [0.53–2.49]; p = 0.718) and HGS (OR = 0.59; 95% CI [0.31–1.16]; p = 0.125).

Conclusions

We concluded that endothelial dysfunction is a key determinant of sarcopenia in KT recipients. Furthermore, endothelial dysfunction is more closely related to gait speed than muscle mass and strength.

Introduction

Sarcopenia, characterized by a progressive decline of skeletal muscle mass, strength, and physical performance, is highly prevalent in patients undergoing dialysis, which leads to poor clinical outcomes (Domanski & Ciechanowski, 2012; Fahal, 2014; Moorthi & Avin, 2017). While kidney transplantation (KT) serves as the optimal treatment for end-stage renal disease, which improves the quality of life and lowers the mortality rate, the prevalence of sarcopenia remains higher than the general population (Limirio et al., 2019; Kosoku et al., 2020). Chronic inflammation, reduced energy and protein intake, nutrient loss, oxidative stress, insulin resistance, and pre-existing comorbidities are well-established risk factors associated with sarcopenia (Fahal, 2014; Stenvinkel et al., 2016). Beyond these factors, vascular dysfunction is postulated to play a crucial role in the pathogenesis of sarcopenia in the rationale of high vascularization of skeletal muscle (Larsson et al., 2019).

As a key regulator of vascular homeostasis, endothelial function may be implicated in the skeletal muscle health. In chronic heart failure patients, impaired endothelial function was more frequently observed in those with sarcopenia (Dos Santos et al., 2017). In community-dwelling older women, there was a significant relationship between endothelial function and skeletal muscle strength (Yoo et al., 2018). While abnormality in endothelial function is highly prevalent in KT recipients (Yildirim et al., 2015), research over the past decade has mainly focused its role on kidney graft survival and cardiovascular outcomes ( Shoskes & Halloran, 1996; Abedini et al., 2010; Frenay et al., 2015); there are no previous studies addressing the association of endothelial dysfunction and sarcopenia in KT recipients.

As a hallmark of endothelial dysfunction, impaired endothelial-dependent dilatation could be assessed by digital thermal monitoring (DTM), a reliable noninvasive method to measure vascular reactivity index (VRI). Poor VRI is well-documented to be correlated with atherosclerosis and predict cardiovascular risks (Naghavi et al., 2016). Thus, the objective of the present study was to explore, in KT recipients, the association of VRI with sarcopenia and its individual components.

Materials & Methods

Participants

This cross-sectional study was conducted in the renal transplant outpatient clinic of the medical center in Hualien, Taiwan. Prevalent KT recipients receiving transplantation for more than 6 months were recruited between September 2015 and March 2016. This study was approved by the Protection of Human Subjects Institutional Review Board of Tzu-Chi University and Hospital (IRB104-84-B). All enrolled patients provided written informed consent and the local ethical committee before study entry. Exclusion criteria were any KT recipients with acute infection within 3 months, acute transplant rejection status, decompensated heart failure, and malignancy at the time of enrollment, or if they refused to provide informed consent for the study.

The basic characteristics, comorbidities, and medications usage were collected by a review of medical records. Comorbid diseases included diabetes mellitus (DM), hypertension, hyperlipidemia, and cardiovascular (CV) disease, which comprised coronary artery disease, myocardial infarction, arrhythmias, or congestive heart failure. The immunosuppressive drug use history included the use of tacrolimus, mycophenolate mofetil (MMF), steroid, rapamycin, and cyclosporine.

Blood pressure and endothelial function measurements

Blood pressure was measured using standard mercury sphygmomanometers after 10-min rest. Endothelial function was assessed by DTM (Endothelix Inc., Houston, TX, USA), which is highly reproducible, fully automated, and non-operator dependent. The measurements were performed after an overnight fast of at least 10 h. After placing the blood pressure cuffs over both bare upper arms and fixing the skin temperature sensors to both of the patient’s index fingers, DTM of both hands was performed during 3 min of stabilization, 2 min of cuff inflation to 50 mmHg greater than systolic blood pressure, and 5 min of deflation. When the cuff was released, blood flow rushed into the forearm and hand, causing a temperature rebound in the fingertip, which was directly proportional to the reactive hyperemia response. During the reactive hyperemia period, the VENDYS software calculated VRI, which assessed the maximum difference between the observed temperature rebound curve and the zero-reactivity curve (Schier et al., 2013; Naghavi et al., 2016). The coefficient of repeatability of temperature rebound and area under the curve previously reported were 2.4% and 2.8% (Ahmadi et al., 2011). Vascular reactivity was classified as being indicative of poor (VRI < 1.0), intermediate (1.0 ≤ VRI < 2.0), or good (VRI ≥ 2.0) vascular reactivity (Naghavi et al., 2016).

Skeletal muscle mass, handgrip strength, and usual gait speed measurements

Bodyweight was measured with the patients wearing light indoor clothing and body height with the patient standing barefoot or stockings. BMI was calculated as weight/height2 (kg/m2). A single-frequency bioimpedance device (Tanita BC 706DB, Tanita Corporation, Tokyo, Japan) was used for the measurement of skeletal muscle mass. Skeletal muscle index (SMI) was calculated as skeletal muscle mass/height2 (kg/m2). Body fat mass was measured using the same device. Low SMI was defined as SMI < 16.5 kg/m2 in men and < 14.2 kg/m2 in women, according to the SMI less than 10% of the sex-specific reference values from Chinese adults (Jin et al., 2019).

Handgrip strength (HGS) was assessed using a Jamar Plus Digital Hand Dynamometer (SI Instruments Pty Ltd, Hilton, Australia), with a precision of 1 kilogram (kg). All patients were instructed to hold the dynamometer in both hands and to squeeze it as hard as they could, with the elbow flexed at 90 degrees, in an upright standing position to keep their arms at their sides. The measurement was repeated three times in both arms with a rest of 1 min before the next measure, and the maximum value was adopted for further analysis. Low handgrip strength was classified as HGS < 28 kg for men and < 18 kg for women, according to the criteria of Asia Working Group for Sarcopenia (Chen et al., 2020). Usual gait speed (GS) was measured by walking for 6 meters at a comfortable pace. Slow GS was defined as GS below 1.0 m/s in all patients (Chen et al., 2020). The same trained operator conducted all procedures.

Definition of sarcopenia

Diagnoses of sarcopenia in our participants were based on the presence of low muscle mass as an essential criterion, accompanied by either low HGS or slow GS ( Cruz-Jentoft et al., 2019; Chen et al., 2020).

Biochemical investigations

From each patient, we collected approximately 5 ml of fasting blood samples. About 1 ml for hemoglobin (Hb) (Sysmex K-1000; Sysmex American, Mundelein, IL, USA) and 4 ml for the others were promptly centrifuged at 3,000 g for 10 min. Serum levels of creatinine, fasting glucose, total cholesterol (TCH), triglycerides, and phosphate were determined by using an auto-analyzer (Siemens Advia 1800; Siemens Healthcare GmbH, Henkestr, Germany). The estimated GFR was calculated based on the CKD-EPI Creatinine equation (Inker et al., 2012). Serum levels of intact parathyroid hormone (PTH) were measured by an autoanalyzer (Diagnostic Systems Laboratories, Webster, Texas, USA).

Statistical analysis

Assuming a 10% prevalence of sarcopenia in our KT recipients and a mean VRI difference of 0.5 between sarcopenia and non-sarcopenia (standard deviation 0.5 in both groups), with an alpha level of 0.05, a total of at least 90 patients should be enrolled to achieve a power of 80%.

The continuous data distribution was examined by using the Kolmogorov–Smirnov test. Variables with normal distribution were expressed as mean ± standard deviation and analyzed by the Student’s independent t-test or Analysis of Variance (ANOVA) test, whereas those not normally distributed were expressed as medians (interquartile ranges) and analyzed by the Mann–Whitney U test or Kruskal-Wallis test. Categorical variables were expressed as number (%) and analyzed by Chi-square test. Multiple comparisons between VRI categories were made by Bonferroni test. Univariate and multivariate logistic regression was used to determine if VRI is associated with sarcopenia and its individual components. In addition to age and gender, the variables with significant differences between sarcopenia and non-sarcopenia groups were regarded as potential confounders and were adjusted in the multivariate models. A p-value of less than 0.05 (two-tailed) was considered statistically significant. The statistical analysis was performed with SPSS 19.0 software (SPSS, Chicago, IL, USA).

Results

A total of 95 patients were enrolled in the study, which consisted of 46 males (48.4%) and 49 females (51.6%). The mean age of the patients was 45.2 ± 10.9 years, and the median time from transplantation was 72.0 months (interquartile range 27.0–111.0). The prevalences of DM, HTN, hyperlipidemia, and CV disease were 45.3%, 38.9%, 41.1%, and 17.9%, respectively. Among them, 48 (50.5%), 34 (35.8%), and 13 (13.7%) of them were classified as having good, intermediate, and poor vascular reactivity, respectively.

The distribution of low SMI, low HGS, slow GS, and sarcopenia overall and by different VRI categories are depicted in Fig. 1. Overall, the prevalence of low SMI, low HGS, and slow GS was 33.7%, 22.1%, and 36.8% in our KT recipients. Sarcopenia was identified in 11.6% of all patients. Notably, among the three VRI groups, the prevalence of low SMI, low HGS, slow GS, and sarcopenia were all higher in patients with VRI < 1.0.

Figure 1 Distribution of low SMI, HGS, slow GS, and sarcopenia in 95 kidney transplantation recipients, all and stratified by VRI categories.

Error bars represent 95% confidence interval. VRI, vascular reactivity index; SMI, skeletal muscle index; HGS, handgrip strength; GS, gait speed.

The baseline characteristics of all patients, with and without sarcopenia, are showed in Table 1. Patients with sarcopenia were lower in BMI (p = 0.001) and VRI (p = 0.041), and have a higher proportion of low muscle mass (p < 0.001), low HGS (p < 0.001), and slow GS (p = 0.001). There were no differences between patients with and without sarcopenia in terms of age, gender, time from transplantation, smoking status, blood pressure, laboratory data, the presence of comorbidities, and the use of medications.

Table 1 Demographic and clinical characteristics of 95 kidney transplantation recipients, with and without sarcopenia.

Characteristics	All
(n = 95)	Sarcopenia
(n = 11)	Non-sarcopenia
( n = 84)	p	
Demographics					
Age (years)	45.2 ± 10.9	44.1 ± 11.6	45.3 ± 10.8	0.724	
Gender, n (%)					
Male	46 (48.4)	6 (54.5)	40 (47.6)	0.666	
Female	49 (51.6)	5 (45.5)	44 (52.4)	
Time from transplantation (months)	72.0 (84.0)	61.0 (120.0)	72.0 (75.0)	0.789	
Current smoking, n (%)	7 (7.4)	0 (0)	7 (8.3)	0.320	
Examination					
SBP (mmHg)	144.0 ± 18.7	138.4 ± 17.9	144.7 ± 18.8	0.292	
DBP (mmHg)	84.1 ± 12.5	80.4 ± 10.3	84.6 ± 12.7	0.296	
VRI	1.9 ± 0.8	1.4 ±0.8	2.0 ±0.8	0.041*	
BMI (kg/m2)	23.3 (5.8)	20.6 (3.5)	24.1 (5.5)	0.001*	
Body fat (%)	29.1 ± 9.2	25.2 ± 8.6	29.6 ± 9.2	0.133	
Low SMI, n (%)	32 (33.7)	11 (100.0)	21 (25.0)	<0.001*	
Low HGS, n (%)	21 (22.1)	7 (63.6)	14 (16.7)	<0.001*	
Slow GS, n (%)	35 (36.8)	9 (81.8)	26 (31.0)	0.001*	
Laboratory data					
Hb (g/L)	120 (36)	120 (40)	120 (40)	0.487	
TCH (mmol/L)	4.77 (1.32)	4.74 (1.50)	4.77 (1.24)	0.629	
Triglyceride (mmol/L)	1.39 (1.05)	1.06 (1.16)	1.41 (1.06)	0.250	
Glucose (mmol/L)	5.33 (1.33)	5.22 (2.39)	5.33 (1.29)	0.439	
eGFR (mL/min/1.73m2)	59.0 ± 26.6	63.9 ± 32.4	58.3 ± 25.9	0.513	
Phosphate (mmol/L)	1.07 ± 0.26	1.07 ± 0.26	1.07 ± 0.26	0.913	
Intact PTH (pmol/L)	10.24 (10.86)	8.87 (15.97)	10.27 (10.52)	0.981	
Diseases, n (%)					
DM	43 (45.3)	4 (36.4)	39 (46.4)	0.528	
HTN	37 (38.9)	4 (36.4)	33 (39.3)	0.852	
Hyperlipidemia	39 (41.1)	5 (45.5)	34 (40.5)	0.752	
CV disease	17 (17.9)	3 (27.3)	14 (16.7)	0.388	
Medications, n (%)					
Tacrolimus	64 (67.4)	8 (72.7)	56 (66.7)	0.687	
MMF	61 (64.2)	5 (45.5)	56 (66.7)	0.168	
Steroid	78 (82.1)	8 (72.7)	70 (83.3)	0.388	
Rapamycin	9 (9.5)	2 (18.2)	7 (8.3)	0.294	
Cyclosporine	17 (17.9)	1 (9.1)	16 (19.0)	0.418	
Notes:

The continuous variables with normal distribution were expressed as mean ± standard deviation, whereas those not normally distributed were expressed as medians (interquartile ranges).

SBP, systolic blood pressure; DBP, diastolic blood pressure; VRI, vascular reactivity index; BMI, body mass index; SMI, skeletal muscle index; HGS, handgrip strength; GS, gait speed; Hb, hemoglobin; TCH, total cholesterol; eGFR, estimated glomerular filtration rate; PTH, parathyroid hormone; DM, diabetes mellitus; HTN, hypertension; CV, cardiovascular; MMF, mycophenolate mofetil.

* p < 0.05 was considered statistically significant.

The clinical characteristics of the 95 KT recipients with good, intermediate, or poor vascular reactivity are presented in Table 2. Patients with poor vascular reactivity have a higher proportion of sarcopenia (p = 0.005), low HGS (p = 0.006), and slow GS (p = 0.029). There was no significant difference in age, gender, time from transplantation, smoking status, blood pressure, laboratory data among the three groups. There were also no differences in the distribution of comorbidities or the use of medications.

Table 2 Clinical characteristics of the study population stratified by vascular reactivity index categories.

Characteristics	Vascular reactivity index	p	
Good (n = 48)	Intermediate (n = 34)	Poor (n = 13)	
Demographics					
Age (years)	44.9 ± 10.8	44.7 ± 11.1	47.5 ± 10.9	0.721	
Gender, n (%)					
Male	21 (43.8)	17 (50.0)	8 (61.5)	0.509	
Female	27 (56.3)	17 (50.0)	5 (38.5)	
Time from transplantation (months)	65.0 (77.0)	86.5 (64.0)	27.0 (63.0)	0.061	
Current smoking, n (%)	2 (4.2)	3 (8.8)	2 (15.4)	0.359	
Examination					
SBP (mmHg)	142.7 ± 18.6	145.0 ± 20.5	146.1 ± 14.3	0.784	
DBP (mmHg)	85.1 ± 12.7	82.8 ± 13.2	83.8 ± 9.8	0.723	
VRI	2.5 ± 0.5	1.6 ± 0.3a	0.4 ± 0.3b,c	<0.001*	
BMI (kg/m2)	23.9 (7.1)	23.6 (3.9)	21.1 (6.5)	0.204	
Body fat (%)	29.4 ± 10.0	30.2 ± 7.6	25.3 ± 9.7	0.265	
Sarcopenia, n (%)	4 (8.3)	2 (5.9)	5 (38.5)b,c	0.005*	
Low SMI, n (%)	17 (35.4)	8 (23.5)	7 (53.8)	0.135	
Low HGS, n (%)	13 (27.1)	2 (5.9)a	6 (46.2)c	0.006*	
Slow GS, n (%)	14 (29.2)	12 (35.3)	9 (69.2)b	0.029*	
Laboratory data					
Hb (g/L)	120 (37)	116 (40)	130 (40)	0.434	
TCH (mmol/L)	4.73 (1.31)	4.82 (1.51)	4.74 (1.26)	0.540	
Triglyceride (mmol/L)	1.29 (1.07)	1.44 (0.99)	1.72 (1.06)	0.583	
Glucose (mmol/L)	5.27 (1.00)	5.55 (2.22)	5.22 (1.50)	0.577	
eGFR (mL/min/1.73m2)	61.5 ± 25.4	55.1 ± 29.3	59.7 ± 23.6	0.570	
Phosphate (mmol/L)	1.07 ± 0.26	1.10 ± 0.29	1.07 ± 0.23	0.815	
Intact PTH (pmol/L)	10.05 (9.47)	10.84 (11.17)	10.60 (15.19)	0.965	
Diseases, n (%)					
DM	20 (41.7)	18 (52.9)	5 (38.5)	0.521	
HTN	17 (35.4)	15 (44.1)	5 (38.5)	0.728	
Hyperlipidemia	17 (35.4)	15 (44.1)	7 (53.8)	0.440	
CV disease	9 (18.8)	6 (17.6)	2 (15.4)	0.960	
Medications, n (%)					
Tacrolimus	32 (66.7)	24 (70.6)	8 (61.5)	0.830	
MMF	31 (64.6)	22 (64.7)	8 (61.5)	0.977	
Steroid	40 (83.3)	28 (82.4)	10 (76.9)	0.866	
Rapamycin	3 (6.3)	5 (14.7)	1 (7.7)	0.424	
Cyclosporine	11 (22.9)	3 (8.8)	3 (23.1)	0.227	
Notes:

The continuous variables with normal distribution were expressed as mean ± standard deviation, whereas those not normally distributed were expressed as medians (interquartile ranges).

Vascular reactivity was classified as being indicative of poor (VRI < 1.0), intermediate (1.0 ≤ VRI < 2.0), or good (VRI ≥ 2.0) vascular reactivity.

a p < 0.05, comparison between intermediate and good VRI.

b p < 0.05, comparison between poor and good VRI.

c p < 0.05, comparison between poor and intermediate VRI.

SBP, systolic blood pressure; DBP, diastolic blood pressure; VRI, vascular reactivity index; BMI, body mass index; SMI, skeletal muscle index; HGS, handgrip strength; GS, gait speed; Hb, hemoglobin; TCH, total cholesterol; eGFR, estimated glomerular filtration rate; PTH, parathyroid hormone; DM, diabetes mellitus; HTN, hypertension; CV, cardiovascular; MMF, mycophenolate mofetil.

* p < 0.05 is considered as statistically significant.

Table 3 demonstrated the univariate and multivariate logistic regression of sarcopenia among KT recipients. In addition to BMI, patients in the poor VRI group were significantly associated with sarcopenia (odds ratio, OR = 6.88; 95% confidence interval CI [1.51–31.29]; p = 0.013), comparing to those with good VRI. This association remained unchanged after adjustment (OR = 6.17; 95% CI [1.06–36.04]; p = 0.043). However, while treating VRI as a continuous variable, the association between VRI and sarcopenia didn’t achieve significance after adjustment (OR = 0.46; 95% CI [0.20–1.04]; p = 0.062).

Table 3 Univariate and multivariate logistic regression analysis of factors associated with sarcopenia among 95 kidney transplantation recipients.

Variables	Unadjusteda	Adjustedb	
Odds ratio (95% CI)	p value	Odds ratio (95% CI)	p value	
VRI entered as a categorical variable	
BMI (kg/m2)	0.64 [0.47–0.87]	0.004*	0.62 [0.43–0.91]	0.014*	
VRI groups	
Good	1 (reference)	1 (reference)	
Intermediate	0.69 [0.12–3.99]	0.688	1.22 [0.17–8.57]	0.845	
Poor	6.88 [1.51–31.29]	0.013*	6.17 [1.06–36.04]	0.043*	
VRI entered as a continuous variable	
BMI (kg/m2)	0.64 [0.47–0.87]	0.004*	0.62 [0.41–0.86]	0.006*	
VRI	0.44 [0.20–0.98]	0.044*	0.46 [0.20–1.04]	0.062	
Notes:

a Nagelkerke’s R-squared in the unadjusted model was 0.264 for BMI, 0.158 and 0.085 for VRI entered as a categorical and continuous variable, respectively.

b Age, gender, BMI, and VRI were adopted in the adjusted model. Nagelkerke’s R-squared in the adjusted model was 0.369 when VRI was entered as a categorical variable and 0.357 when treating VRI as a continuous variable.

BMI, body mass index; VRI, vascular reactivity index.

* p < 0.05 is considered as statistically significant.

Table 4 depicted the associations of VRI with low SMI, HGS, and slow GS individually. After full adjustment, VRI entered as a continuous variable independently predicted slow GS significantly (OR = 0.41; 95% CI [0.21–0.79]; p = 0.007), but not low HGS (OR = 0.59; 95% CI [0.31–1.16]; p = 0.125) and SMI (OR = 1.15; 95% CI [0.53–2.49]; p = 0.718). Similar results were achieved when VRI was entered as a categorical variable.

Table 4 Multivariate logistic regression analysis of the associations of VRI with low SMI, HGS, and slow GS in 95 kidney transplantation recipients.

Variables	Low SMI	Low HGS	Slow GS	
Odds ratio
(95% CI)	p value	Odds ratio
(95% CI)	p value	Odds ratio
(95% CI)	p value	
VRI entered as a categorical variable	
VRI groups	
Good	1 (reference)	1 (reference)	1 (reference)	
Intermediate	0.88 [0.24–3.23]	0.850	0.17 [0.03–0.81]	0.026*	1.42 [0.53–3.80]	0.481	
Poor	1.87 [0.25–14.12]	0.544	2.67 [0.70–10.23]	0.152	7.30 [1.73–30.83]	0.007*	
VRI entered as a continuous variable	
VRI	1.15 [0.53–2.49]	0.718	0.59 [0.31–1.16]	0.125	0.41 [0.21–0.79]	0.007*	
Notes:

Age, gender, BMI, and VRI were adjusted in the models.

VRI, vascular reactivity index; SMI, skeletal muscle index; HGS, handgrip strength; GS, gait speed.

* p < 0.05 is considered as statistically significant.

Discussion

The primary findings of the present study are that lower BMI and poor VRI were two independent factors associated with sarcopenia in our KT recipients. Furthermore, among the three components of sarcopenia, VRI appeared to predict GS, but no SMI and HGS.

Few previous studies investigated the link between endothelial dysfunction and sarcopenia and its individual components. In chronic heart failure patients, those with sarcopenia had a lower baseline and peak reactive hyperemia blood flow in the forearm and leg than those without sarcopenia. This reactive hyperemia was closely associated with physical performance, included peak VO2 and 6-min walk distance (Dos Santos et al., 2017). In community-dwelling older women, endothelial dysfunction independently predicted handgrip strength weakness (Yoo et al., 2018). Elevated serum levels of asymmetric dimethylarginine, a surrogate biomarker of endothelial dysfunction, were not only associated with lower muscle strengths and slower GS among elderly individuals (Obayashi et al., 2016) but also with reduced physical performance in the 10-repetition sit-to-stand test in prevalent hemodialysis patients (Pajek et al., 2018). In accordance with aforementioned findings, our study, for the first time, showed a significant association between endothelial dysfunction and sarcopenia in KT recipients.

Skeletal muscle mass, HGS, and GS, the three components of sarcopenia, may have different clinical relevance. Among our KT recipients, we interestingly found that VRI was more closely related to GS than HGS and SMI. This finding indicated that endothelial dysfunction may have more adverse impacts on physical performance than on muscle mass and strength. Physiologically, the performance of gait speed test is more dependent upon enhanced local blood flow, compared to the production of maximal handgrip strength. Consistent with our finding, in elderly and patients with pre-dialysis chronic kidney disease, endothelial dysfunction is associated with frailty, a clinical syndrome characterized by a decline in physiological reserve and poor physical performance (Alonso-Bouzón et al., 2014; Mansur et al., 2015).

Several possible mechanisms explained the crosslink between endothelial dysfunction and sarcopenia in KT recipients. First, skeletal muscle microcirculation plays a vital role in skeletal muscle health (Hendrickse & Degens, 2019). Endothelial dysfunction was shown to impair microcirculation of skeletal muscle and downregulate vascular endothelial growth factor expression, which compromises angiogenesis and hampers muscle regeneration (Latroche et al., 2015). In vitro study, treatment of asymmetric dimethylarginine, an endothelial inhibitor of nitric oxide synthesis, results in increased protein degradation in cultured C2C12 myotubes (Zhou et al., 2009). Second, impaired endothelial function in skeletal muscle may reduce the delivery of dietary amino acids to skeletal muscle fibers and impair skeletal muscle protein synthesis (Moro et al., 2016). Third, emerging evidence showed myokines, which are hundreds of cytokines and proteins secreted by skeletal muscle in response to muscle contraction, exert autocrine, paracrine, and endocrine effects on various systems, including the vascular system (Severinsen & Pedersen, 2020). For example, follistatin-related protein one, a muscle-secreted glycoprotein, has been shown to regulate endothelial cell function and blood vessel growth in skeletal muscle through a nitric-oxide synthase-dependent mechanism (Ouchi et al., 2008). Fourth, endothelial dysfunction compromises cardiovascular health, which may manifest as muscle weakness and poor physical performance. Finally, endothelial dysfunction and sarcopenia shared some common pathogenic mechanisms, such as inflammation, insulin resistance, and oxidative stress, which could explain the close link to each other (Fahal, 2014; Goligorsky, 2015).

Apart from endothelial dysfunction, BMI was another relevant factor strongly associated with sarcopenia in our KT recipients. Consistent with our findings, Kosoku et al. showed that BMI is a reliable nutritional marker for predicting sarcopenia in KT recipients, which has a good discrimination performance (Kosoku et al., 2020).

Surprisingly, aging is recognized as an irrefutable predictor for sarcopenia in the geriatric population (Yamada et al., 2013; Han et al., 2016), but not in our study. This discrepancy may be explained by the relatively young age of our enrolled participants.

This is the first study to report the association between endothelial dysfunction and sarcopenia in KT recipients. However, several significant limitations need to be considered. First, the total sample size was relatively small, and a small proportion was categorized as having sarcopenia and poor VRI, which could affect the precision of estimation. Second, we directly measured vascular reactivity but didn’t measure some biomarkers of endothelium dysfunction, such as asymmetric dimethylarginine and Soluble Vascular Cell Adhesion Molecule-1. Third, a bioimpedance device was used for the measurement of skeletal muscle mass. Computed tomography and magnetic resonance imaging, the gold standard for the assessment of skeletal muscle mass, were not used. Fourth, inflammatory markers, such as C-reactive protein and interleukin-6, were not available in this study. Fifth, the causal relationships between endothelial function and sarcopenia couldn’t be established by the cross-sectional design. Finally, the participants in this study were relatively young, and our results may not be generalizable to elderly KT recipients.

Conclusions

Our study showed a significant association between endothelial dysfunction and sarcopenia in KT recipients, which implicated that endothelial dysfunction may be involved in the pathogenesis of sarcopenia. Furthermore, endothelial dysfunction is more closely related to physical performance than muscle mass and strength. Further longitudinal studies investigating the changes among endothelial function, skeletal muscle mass, and function are warranted to establish the causal relationship.

Supplemental Information

Supplemental Information 1 Raw data.

Click here for additional data file.

Supplemental Information 2 Codebook.

Click here for additional data file.

Additional Information and Declarations

Competing Interests

Author Contributions

Human Ethics

Data Availability

Bang-Gee Hsu is an Academic Editor for PeerJ.

Siok-Bin Khoo performed the experiments, authored or reviewed drafts of the paper, and approved the final draft.

Yu-Li Lin conceived and designed the experiments, analyzed the data, prepared figures and/or tables, authored or reviewed drafts of the paper, and approved the final draft.

Guan-Jin Ho performed the experiments, analyzed the data, prepared figures and/or tables, and approved the final draft.

Ming-Che Lee conceived and designed the experiments, authored or reviewed drafts of the paper, and approved the final draft.

Bang-Gee Hsu conceived and designed the experiments, authored or reviewed drafts of the paper, and approved the final draft.

The following information was supplied relating to ethical approvals (i.e., approving body and any reference numbers):

This study was approved by the Protection of Human Subjects Institutional Review Board of Tzu-Chi University and Hospital (IRB104-84-B).

The following information was supplied regarding data availability:

The raw data are available in the Supplementary File.

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
