# Peer review of "Association of endothelial dysfunction with sarcopenia and muscle function in a relatively young cohort of kidney transplant recipients"

_PeerJ, doi:10.7717/peerj.12521_

## Round 0.1 · original submission · Major Revisions

I apologise for the delay in getting a decision to you on your submission.
On this occasion, it has taken much longer than usual to secure two informative and independent reviews. The reviewers' have made several constructive suggestions which will help improve your manuscript.

Although many aspects of manuscript are well reported and appear technically sound, both reviewers have raised concerns about the sample size of study and the very small number of patients with sarcopenia (n=11) on which multivariate analysis has been performed. I concur with this assessment, and while you acknowledge this as a limitation is does not mitigate this as an issue.

The model presented in Table 3 shows inclusion of 10 different variables, including several categorical ones. The results from such an analysis are unlikely to be very robust with such small number of cases of interest. Indeed, it is not clear why these specific variables were included inn the model, as according to Tables 1 & 2, eight of them were not associated with either sarcopenia or VRI, respectively. In the text (line 204) you describe “adjusting for confounders” but these variables cannot confound the relationship with sarcopenia and VRI because they are not associated with either. Please clarify the purpose of this modelling and consider carefully whether such adjustment is appropriate. I would consider consulting with a statistician if you are uncertain of how to proceed.

How well does your model fit the observed outcome? What is the R-squared value and how much of this variation is explained by VRI and BMI?

In addition to these major concerns, please attend to the following issues:

1. Comment on the reproducibility of VRI measurements in your hands
2. Report biochemical values in SI units (e.g., mmol/L is preferred to mg/dl)
3. Report eGFR rather than creatinine
4. Use “phosphate” not phosphorus
5. Refrain from comments about “marginally” significant results e.g. line 208. You define statistical significance as p<0.05 (presumably two -tailed?). If the P value is equal to or greater than 0.05 it is not significant.


·

Basic reporting

The paper is well written throughout, with a clear background provided in the introduction that allows the reader to understand the context to which the work is relevant.

specific points:

Title
The authors might consider amending the title of the paper to acknowledge the relatively young cohort of patients included in the study. Furthermore, I think it would be beneficial to refer to ‘muscle function’ in the title, to better characterise the work.

Abstract
‘Kg’ should be ‘kg’

Introduction
Line 64: it would be useful to include a citation to back up this statement, there a plenty of studies on the potential role of the muscle microcirculation/endothelial dysfunction (see Larsson et al 2019 Phys Rev).

Line 66-68: ‘cardiovascular events’ implies pathological cardiac incidents/performance rather than endothelial dysfunction, per se.

Table 1
Add in a line on the patient data from men.

Inclusion of brackets and ‘±’ is used interchangeably throughout; if this is deliberate then the authors should make clear the significance of this distinction.

Figure 1
Rather than noting the percentage values above each bar, error bars should be included to enable the variation in each sample to be taken into account.

Figure 2
I think that this figure would be better presented in a table similar to Table 3.

Experimental design

Methods
Make clear why the unadjusted and adjusted multivariate approaches are presented in the results.

Validity of the findings

Results
It would be interesting to explore the interaction of BMI, VRI and sarcopenia, because it would seem that these sarcopenic patients have consistently low BMI. If you consider variation in the multivariate model after removal of BMI (which accounts for the greatest proportion according to p-values in Table 3), I wonder if any of the other covariates become statistically significant? It strikes me that BMI will covary closely with SMI due to the muscle loss that drives decreases in both indices, so BMI is perhaps not an independent predictor but rather a derived measure of sarcopenia, in which case it might be obscuring other (currently non-significant) factors in the multivariate model.

Line 204: the univariate models indicated that these factors were not significantly related to the development of sarcopenia, so cannot strictly be cited as ‘confounding’ factors; no evidence of any test on interactions between dependent variables is presented in the multivariate model, so it is difficult to envisage why they would account for more of the variation (in the independent variable) after inclusion of more covariates.

Line 208: I appreciate that the P-value for handgrip strength is close to the 5% significance level, but by the statistical definition provided in the manuscript, this P-value remains non-significant, marginal or not.

Discussion

Line 213: same as line 208, the result was non-significant; I interpret this from a physiological point of view – the handgrip protocol will elicit rapid muscle force production that does not depend upon enhanced local blood flow whereas the gait score protocol likely does, and so endothelial reactivity is potentially more important for this sustained exercise.

Additional comments

I think that the authors have produced an interesting study on the incidence of endothelial dysfunction in the development of, and/or coincidence with, sarcopenia. Overall this paper is well-written with a clearly defined objective. I think the authors should consider my comments to improve the focus and clarity of the manuscript.

·

Basic reporting

It was a well written article, with adequate english, literature references and background.
The article structure, figures and tables are well done. And, the raw data shared is organized, well described.
I appreciated the hypotheses and results.
However, as I have no experience with the terms, methods and clinic of the cardiovascular area, I believe that a second reviewer with a background in cardiovascular disease could enrich the review.

Experimental design

Introduction, well adressed the content.

About methods I have only one question. There is no sample size calculation, therefore the authors are not authorized to say this is either a small or a large sample, as it was written in the last paragraph of discussion section.
In fact, this is the basic essential tenet of any research question, and when not estimated or reached, it is a huge limitation of the conclusions. Sample size insufficiency threatens the validity and generalizability of studies’ results. The size of a sample influences two statistical properties: 1) the precision of estimates and 2) the power of the study to draw conclusions.
So, having said so, none of the statistics can support the conclusions, unless the authors provide this fundamental, basic methodological scientific requirement. Otherwise, they must discuss why they have not carried it out and discuss as a huge limitation of the study...

Validity of the findings

About discussion section:
- Line 213: According to the results, VRI appear to predict only GS. HGS and SMI had a p value >0.05, so, VRI do not predicted both.
Limitation: add as limitation of the study the limitations of body composition evaluated by BIA.

---

## Round 0.2 · accepted · Accept

Many thanks for your detailed response and appropriate revisions to the manuscript.

·

Basic reporting

no comment

Experimental design

no comment

Validity of the findings

no comment

Additional comments

The authors have carefully revised the manuscript, and I think that it is improved as a consequence. Therefore I have no further comments on the paper.